# Slow-growing *Salmonella enterica* serovariety Typhi mis-identified as serovariety Gallinarum in Ibadan, Nigeria

Gabriel Temitope Sunmonu[1], Veronica O. Ogunleye[2], Odion O. Ikhimiukor[1], Oluwafemi A. Popoola[3], Precious E. Osadebamwen[1], Aderemi Kehinde[2,4], Iruka N. Okeke[1,4]*

1 Department of Pharmaceutical Microbiology, and Biotechnology, Faculty of Pharmacy, University of Ibadan, Ibadan, Nigeria, 2 Department of Medical Microbiology & Parasitology, University College Hospital, Ibadan, Nigeria, 3 Department of Community Medicine, College of Medicine, University of Ibadan, Ibadan, Nigeria, 4 Department of Medical Microbiology & Parasitology, College of Medicine, University of Ibadan, Ibadan, Nigeria

* iruka.n.okeke@gmail.com

## Abstract

*Salmonella enterica* subsp. *enterica* serovar Typhi is endemic in Nigeria where serovar Gallinarum is rarely reported. Following routine blood culture and identification, three patients with probable *S.* Gallinarum infections were reported in Ibadan within 10-days, precipitating an outbreak investigation. All three isolates were re-identified using VITEK2, motility-tested using Sulphide-Indole-Motility (SIM) medium and whole genome sequenced (WGS) on Illumina and Oxford Nanopore platforms. Short reads were used to determine sequence type, serotype and phylogenetic relationships to previously characterized *S.* Typhi from Nigeria. Chromosomal configurations were deduced from hybrid assemblies using socru v2.2.4. The three isolates, initially identified as *S.* Gallinarum, re-identified as gamma glutamyl transferase-positive *S.* Typhi genotype 3.1.1 by VITEK-2 and WGS. Two individuals in the same household yielded isolates with no single nucleotide polymorphisms, suggesting a point-source and the third case was an independent infection event. Motility was negative by hanging drop method and in SIM medium at 24h, but positive after 48h incubation. The three isolates formed smaller colonies than typical *Salmonella* strains. Hybrid genome assemblies revealed chromosomal fragments arrangements with imbalances on either side of *ori* and *ter*, recently been shown to slow *S.* Typhi growth. We have shown that slow-growing *S.* Typhi can be mistaken for *S.* Gallinarum and that a cluster of blood culture-derived *S.* Typhi, with imbalanced chromosome rearrangements, was thereby mistaken for a zoonotic *S.* Gallinarum outbreak. Suspected *S.* Gallinarum isolates in typhoid endemic areas should be evaluated biochemically and for motility after extended incubation, and verified by serological or molecular methods.

**Data availability statement:** The sequence reads generated in this study have been deposited in the NCBI (National Center for Biotechnology Information) database under the following BioSample accession numbers: PRJNA1197410, SAMN45804238 (https://www.ncbi.nlm.nih.gov/biosample/SAMN45804238), SAMN45804239 (https://www.ncbi.nlm.nih.gov/biosample/SAMN45804239), and SAMN45804240 (https://www.ncbi.nlm.nih.gov/biosample/SAMN45804240).

**Funding:** At the time of isolation of the bacterial isolates used in this study, blood culture-based surveillance in Ibadan was supported by the International Vaccine Institute through the SETA-Plus program. This research was funded by the UK National Institute for Health and Care Research (NIHR) Global Health Research Unit on Genomics and Enabling Data for Surveillance of Antimicrobial Resistance (Consortium Award: NIHR Project #NIHR133307). I.N.O. is supported as a Calestous Juma Science Leadership Fellow by the Bill & Melinda Gates Foundation (Grant INV-036234). The funders had no role in study design, data collection and analysis, decision to publish, or preparation of the manuscript.

**Competing interests:** The authors have declared that no competing interests exist.

## Introduction

*Salmonella enterica* infections pose a significant public health concern in Africa, contributing substantially to morbidity and mortality rates across the continent [1]. Nigeria, in particular, wrestles with endemic *Salmonella enterica* subsp. *enterica* serovar Typhi and invasive non-typhoidal *Salmonella* lineages of importance [2,3], which persist due to inadequate sanitation, limited access to clean water, and insufficient food safety measures [4].

*Salmonella enterica* subsp. *enterica* comprises over 2,600 serovars with varying host specificities and clinical manifestations [5]. Historically, these have been classified into typhoidal serovars (*Salmonella* serovar Typhi and *Salmonella* serovar Paratyphi), which are human-adapted and cause enteric fever, and non-typhoidal serovars (NTS) such as *Salmonella enterica* serovar Typhimurium and *Salmonella enterica* Enteritidis, which typically cause self-limiting gastroenteritis in immunocompetent hosts but can lead to invasive disease in immunocompromised individuals [6,7]. In sub-Saharan Africa, invasive non-typhoidal Salmonella (iNTS) disease has emerged as a major public health threat, particularly affecting children under five years of age and individuals with comorbidities such as malaria, HIV infection, and malnutrition [1,8].

*Salmonella enterica* subsp. *enterica* serovar Gallinarum represents a host-adapted serovar that primarily affects avian species, causing fowl typhoid with high mortality rates in poultry flocks [9]. Unlike many other *Salmonella* serovars, *S.* Gallinarum has historically been considered non-zoonotic or minimally pathogenic to humans due to its host adaptation [10]. However, recent evidence suggests that zoonotic transmission to humans can occur, particularly in settings with close human-poultry contact, such as backyard poultry operations common in many African communities [11–13]. Human infections with *S.* Gallinarum typically present as mild gastroenteritis but can occasionally progress to invasive disease, especially in immunocompromised individuals [13,14].

The University College Hospital, Ibadan has considerable expertise isolating and identifying *Salmonella* from blood, being part of the Severe Typhoid in Africa project since 2016 [15]. According to protocol, *Salmonella* is identified using commercial strip biochemicals (API20E). Typhi is confirmed by Vi serology and invasive non-typhoidal *Salmonella* are confirmed by PCR [15,16]. Since surveillance began, *S.* Typhi, the human adapted cause of typhoid fever, has been shown to be endemic in Ibadan [16,17]. By contrast, *S.* Gallinarum, a poultry pathogen that can cause zoonotic infections in humans, has never been detected since 2018 and is rarely reported in Nigeria [16,18]. In November 2021, routine invasive *Salmonella* surveillance in Ibadan detected three "*S.* Gallinarum" isolates within ten days, two from individuals that indicated they had a backyard poultry, precipitating an outbreak investigation and request for reference lab re-identification.

## Methods

### Ethics statement

Ethical approval for this research was received from UI/UCH ethics committee, with approval numbers UI/EC/22/0113 and UI/EC/21/0191. Proposal UI/EC/21/0191.

covers surveillance and genomic analysis of *Salmonella* isolates in Ibadan metropolis whilst proposal UI/EC/22/0113 cover surveillance at all sentinels in Nigeria, including Ibadan. Individuals from whom the isolates were obtained provided written informed consent to participate.

## Strains

Routine blood culture surveillance has been conducted in Ibadan from 1 February 2017 to date. Isolates reported in this study were recruited in in 2021–2022. Three isolates recovered from different patients in Ibadan within 10 days in November 2021(9 Nov 21, 17 Nov 21 and 17 Nov 21) were identified as presumptive *S.* Gallinarum based on routine testing at the University College Hospital (UCH) Ibadan, Nigeria. Address and other metadata were retrieved for each patient. *Salmonella* Typhimurium ATCC 14028, other isolates collected in invasive *Salmonella* surveillance from Ibadan and other sites between 8 Jul 2021 and 4 Dec 2021 (Ibadan) and 4 Jul 2022 and 10 Oct 2022 (other sites) [16] or antimicrobial resistance surveillance [3] were used as controls and are listed in S1 Table in the supporting information. The isolates were cryopreserved in 50% LB:glycerol at -80°C.

## Colony morphology and identification

Presumptive *Salmonella* isolates from patients were initially identified using API20E and determining motility by microscopy using the hanging-drop method. The isolates' colony morphology was observed on nutrient agar. Isolates were re-identified and biotyped with the Gram-negative (GN) test kit (Ref: 21341) on VITEK-2 systems (version 2.0, Marcy-l'Etoile, France, Biomérieux) according to manufacturer's instructions and compared to extended biochemical profiles of other *S.* Typhi isolates. Motility testing, initially negative by hanging-drop method, was repeated using SIM (Sulphide-Indole-Motility) semi-solid medium.

## Growth curves and growth rate determination

The three test isolates, earlier isolated *S.* Typhi – including slow-growers, and *Salmonella* Typhimurium ATCC 14028 were streaked onto nutrient agar plates and incubated for 24 hours. Following incubation, 3–4 pure colonies was selected from the plate and grown overnight in broth. Using sterilized normal saline, the isolates were standardized to 0.5 MacFarland ($1.5 \times 10^8$ cfu), ensuring uniformity in the number of cells inoculated. A 5 μl aliquot of the standardized inoculum was then introduced into a 195 μl liquid medium (Dulbecco's Modified Eagle Medium (DMEM) broth without a colour indicator) in triplicate and incubated on a shaker to promote aeration and homogeneous growth conditions. Optical density (OD) was measured at intervals (0, 1, 2, 4, 6, 8, 10, 16, 24, and 36 hours). For each time point, the OD of the culture was measured using a spectrophotometer (Thermo 96 wells MULTISKAN FC) carrying a 595 nm filter. Absorbance at 595 nm were plotted on a logarithmic axis against time using GraphPad Prism 10. Growth rates were computed Microsoft excel using the formula μ = (ln OD2 – ln OD1)/ (t2 – t1) where μ = bacterial growth rate; t1 (start of log phase) = 2 hrs; t2 (late/ end of log phase) = 10hrs; OD1 = optical density at 2 hrs; OD2 = optical density at 10 hrs.

## DNA extraction, library preparation and whole genome sequencing

For short-read whole-genome sequencing, DNA of the isolates was extracted using Wizard DNA extraction kit (Promega; Wisconsin, USA) in accordance with manufacturer's protocol. The extracted DNA was quantified on a Qubit fluorometer (Invitrogen; California, USA) using a dsDNA Broad Range quantification assay. Using the NEBNext Ultra II FS DNA library kit for Illumina, which has 96 unique indexes (New England Biolabs, Massachusetts, USA; Cat. No. E6609L), dsDNA libraries were prepared. DNA libraries was quantified using dsDNA High Sensitivity quantification assay on a Qubit fluorometer (Invitrogen; California, USA) and fragment length analysed with the Bioanalyzer (Agilent). Denatured libraries were sequenced on an Illumina MiSeq (Illumina, California, USA). The raw sequence reads were *de novo* assembled using SPAdes v3.15.3 [19] according to GHRU protocols (https://gitlab.com/cgps/ghru/pipelines/dsl2/pipelines/assembly).

We performed long-read whole-genome sequencing of the isolates using Oxford Nanopore technology, which allowed us to obtain completely assembled genomes for comprehensive analyses. Genomic DNA of the isolates was extracted using the FastDNA spin kit for soil (MP Biomedicals, Santa Ana, CA, USA) in accordance with manufacturer's protocol to obtain less fragmented DNA. Long-read sequencing libraries were then prepared using the Ligation Sequencing Kit (SQK-LSK109) and sequenced on a MinION Flow Cell (R9.4.1) with MinKNOW version 22.08.9 (Oxford Nanopore Technologies, Inc., Oxford, United Kingdom). We then carried out super accuracy base calling and demultiplexing on the generated reads using MinKNOW version 22.08.9.

### Sequence typing of *Salmonella*

Sequence reads were deposited in the *Salmonella* database for *Salmonella* on EnteroBase [20], which was also used to determine multi-locus sequence types (MLST). The genome assemblies were analysed using the *Salmonella* In-Silico Typing Resource (SISTR) for the prediction of serovars and serogroups [21].

### Single Nucleotide Polymorphism (SNP) calling and phylogenetic analysis

We performed a SNP-based phylogenetic analysis to determine the phylogenetic relationships between the *Salmonella* strains and previously characterized *S.* Typhi from Nigeria [3,16]. Reference sequences for the *S. enterica* genomes were objectively selected from the National Center for Biotechnology Information Reference Sequence (RefSeq) database (https://www.ncbi.nlm.nih.gov/refseq/). The selected reference was *S. enterica* subsp. enterica serovar Typhi strain H12ESR00755-001A (accession number LT905142). The sequence reads for each isolates were then mapped to the chromosome of the reference using BWA (v0.7.17) [22] and variants were called and filtered using bcftools (v1.9) [23]. Variant positions were concatenated into a pseudoalignment and used to generate a maximum likelihood tree using iqtree (v1.6.8) [24]. The phylogenetic tree was visualized and annotated using Interactive Tree of Life (iTOL) v.6.8.1 [25]. SNP distances between the genome pairs were calculated using snp-dists v.0.8.2 (https://github.com/tseemann/snp-dists).

### *Salmonella* chromosomal rearrangement identification

Hybrid assemblies were generated from short and long whole genome reads using the nf-core/bacass 2.1.0 pipeline (https://nf-co.re/bacass/2.1.0), and genome structure assignments were called and visualised using *socru* v2.2.4 [26]. The terminus of replication (*ter*) was annotated relative to the *tus* gene while the origin of DNA replication (*oriC*) was determined using Ori-Finder [27].

### Statistical analysis

Categorical data were compared using a two-tailed Fisher's exact test executed using EpiInfo software downloaded from https://www.cdc.gov/epiinfo/index.html.

## Results

### *Salmonella* identity and biochemical profile

As shown in Fig 1 most *Salmonella* recovered from blood in Ibadan are *Salmonella* Typhi or non-Gallinarum invasive non-typhoidal *Salmonella*. However, three presumptive *Salmonella* Gallinarum isolates were detected in November 2021. Epidemiologic follow up revealed that two isolates were recovered from the same household, which had a backyard poultry, further suggesting that *Salmonella* Gallinarum zoonosis was possible.

Sentinel identification had marked the isolates as likely *S.* Gallinarum, with low discrimination based on API 20E testing, read at 24 hours (Table 1). The API 20E profile was 4004100, with an ID probability of 39% for Gallinarum, with *E coli* type 2 (37% probability), *Hafnia alvei* (22% probability) and *Salmonella* Typhi (1% probability) as possible alternatives. The fact

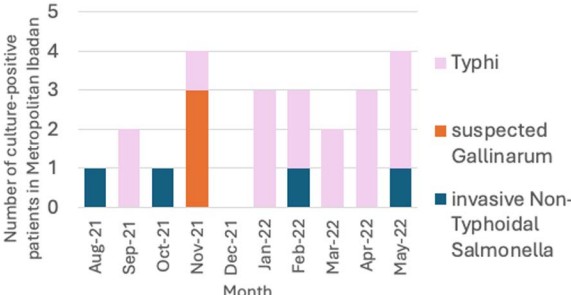

**Fig 1. Recovery of invasive *Salmonella* from blood cultures on patients resident in metropolitan Ibadan (August 2021-May 2022).**

**Table 1. API 20E Profile of the suspected *S.* Gallinarum in this study at 24 h and documented profiles of different Salmonella serovars.**

| API 20 E V4.0 | *S.* Paratyphi A | *S.* Pullorum | *Salmonella spp* | *S.* Arizonae | *S.* Choleraesuis | *S.* Typhi | *S.* Gallinarum | Strains in this study |
|---|---|---|---|---|---|---|---|---|
| ONPG | 0 | 0 | 1 | 98 | 0 | 0 | 0 | – |
| ADH | 5 | 1 | 56 | 75 | 15 | 1 | 1 | – |
| LDC | 0 | 75 | 82 | 97 | 99 | 99 | 100 | + |
| ODC | 99 | 100 | 93 | 98 | 99 | 0 | 1 | – |
| CIT | 0 | 0 | 65 | 75 | 6 | 0 | 0 | – |
| H2S | 1 | 85 | 83 | 99 | 64 | 8 | 25 | – |
| URE | 0 | 0 | 0 | 0 | 0 | 0 | 0 | – |
| TDA | 0 | 0 | 0 | 0 | 0 | 0 | 0 | – |
| IND | 0 | 0 | 1 | 1 | 0 | 0 | 0 | – |
| VP | 0 | 0 | 0 | 0 | 0 | 0 | 0 | – |
| GEL | 0 | 0 | 1 | 0 | 0 | 0 | 0 | – |
| GLU | 100 | 100 | 99 | 100 | 100 | 100 | 100 | + |
| MAN | 99 | 100 | 100 | 99 | 99 | 99 | 100 | + |
| INO | 0 | 0 | 40 | 0 | 0 | 0 | 0 | – |
| SOR | 99 | 0 | 99 | 99 | 98 | 99 | 0 | – |
| RHA | 98 | 100 | 86 | 99 | 99 | 0 | 1 | – |
| SAC | 0 | 0 | 1 | 1 | 0 | 0 | 0 | – |
| MEL | 96 | 0 | 90 | 78 | 20 | 99 | 0 | – |
| AMY | 0 | 0 | 1 | 0 | 0 | 0 | 0 | – |
| ARA | 99 | 75 | 99 | 99 | 0 | 0 | 100 | – |
| MOB | 95 | 0 | 95 | 99 | 95 | 97 | 0 | – |

Abbreviations: ONPG: Ortho-Nitrophenyl-β-galactoside, ADH: Arginine Dihydrolase, LDC: Lysine Decarboxylase, ODC: Ornithine Decarboxylase, CIT: Citrate utilization, H2S: Hydrogen Sulfide production, URE: Urease, TDA: Tryptophan Deaminase, IND: Indole production, VP: Voges-Proskauer, GEL: Gelatinase, GLU: Glucose fermentation, MAN: Mannitol fermentation, INO: Inositol fermentation, SOR: Sorbitol fermentation, RHA: Rhamnose fermentation, SAC: Sucrose fermentation, MEL: Melibiose fermentation, AMY: Amygdalin fermentation, ARA: Arabinose fermentation, MOB: Mobility, -: negative, +: positive. Using the hanging drop technique, our strains were negative for motility.

that epidemiological investigation determined that two of the individuals lived in a household with a backyard poultry further supported possible *S.* Gallinarum zoonosis and the fact that the third individual had no contact with poultry or the other two individuals gave reasons for public health concern. Based on the low discrimination identification and the possibility of a cluster, the isolates were forwarded to the reference lab where, the three isolates, initially identified as *S.* Gallinarum, were re-identified as *S.* Typhi by VITEK-2. The VITEK-2 biochemical profile of the strains in this investigation and other *S.* Typhi

isolates is shown in Fig 2. Of the 49 strains tested, 15 (30.6%) had the bionumber 0005610440104210, which has a probability of 95% of being *S*. Typhi. Five other bioprofiles were identified but there was no profile common to all three strains that identified as *S*. Gallinarum by API 20E. GGT (Gamma Glutamyl Transferase) positivity was significantly more common among three presumptive *S*. Gallinarum that were confirmed as *S*. Typhi than 46 other strains that identified as Typhi originally by API 20E (p = 0.001) (Fig 2) in the 3 strains being investigated which is also present in 2 other *S*. Typhi isolates. Retesting motility using SIM medium revealed that the strains were slowly motile (Table 2).

## Morphology and growth rate

The three Typhi isolates that were initially misidentified as *S*. Gallinarum produce noticeably smaller colonies on nutrient agar plates than most other *Salmonella* bloodstream isolates (Fig 3C), and growth rates were slower (Fig 3A and 3B) compared to *S*. Typhimurium strain ATCC 14028 in liquid media. We were additionally able to discern that although there are Typhi strains that grow at rates comparable to this strain, previously isolated strains ILO ET_19_049 and UCH_SETA_ N3004 grown slower in minimal media. However, the three strains recovered in this study were the slowest growing. They had a prolonged lag phase, lower cell density and reduced growth rate as shown in Fig 3A and 3B.

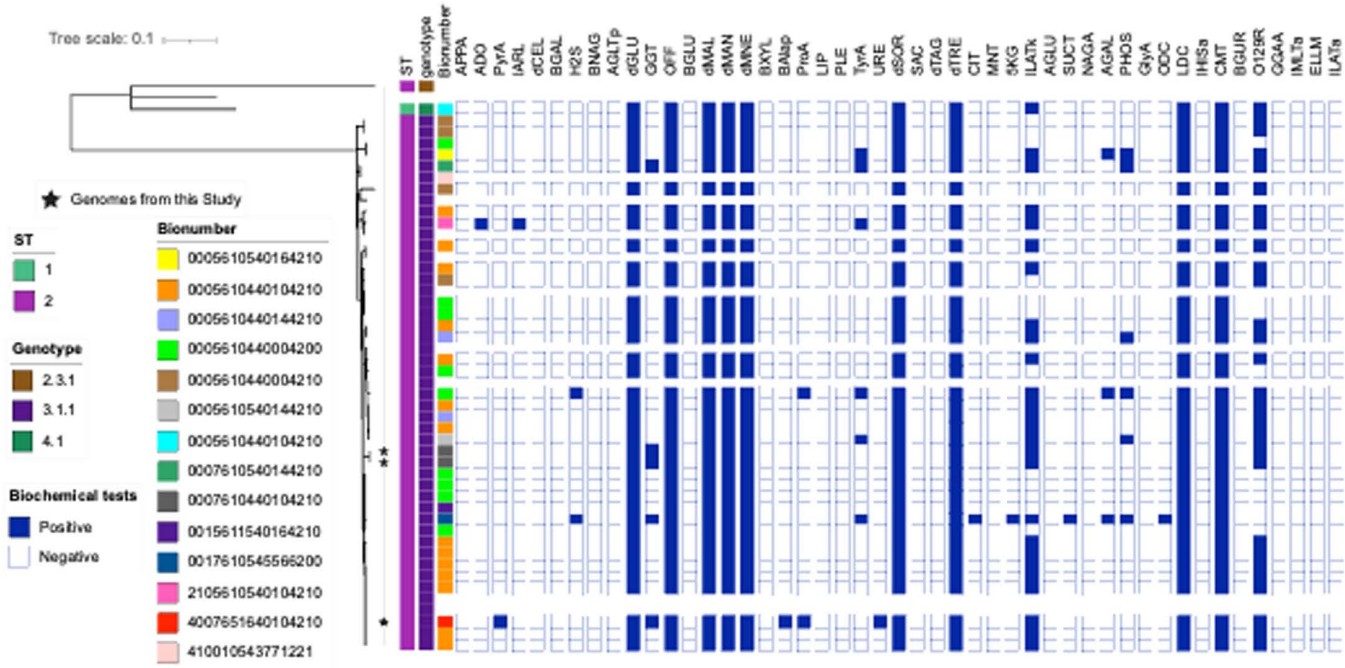

**Fig 2. A mid-point rooted maximum likelihood phylogeny of 46 *S*. Typhi isolated in Ibadan [ 3,16] and 3 genomes sequenced from this study (represented by black stars) showing Sequence Types, genotypes, Bionumbers and VITEK-2 Biochemical Profiles.** The presumptive Gallinarum isolates were of bioprofiles 0007610440104210 (two isolates) and 4007651640104210 [one isolate] with respective *S*. Typhi identification probabilities of 95% (https:// microreact.org/project/7to9opSB7e1ACJkCKyu8bd-sgalstyphi-analysis). The tree was mid-point rooted. Abbreviations: APPA, Ala-Phe-Pro-Arylamidase; ADO, Adonitol; PyrA, L-Pyrrolydonyl-Arylamidase; lARL, L-Arabitol; dCEL, D-Cellobiose; BGAL, BETA-Galactosidase; H2S, H2S Production; BNAG, BETA-N-Acetyl-Glucosaminidase; AGLTp, Glutamyl Arylamidase pNA; dGLU, D-Glucose; GGT, Gamma-Glutamyl-Transferase; OFF, Fermentation/ Glucose; BGLU, Beta-Glucosidase; dMAL, D-Maltose; dMAN, D-MANNITOL; dMNE, D-Mannose; BXYL, BETA-Xylosidase; BAlap, BETA-Alanine arylamidase pNA; ProA, L-Proline Arylamidase; LIP, LIPASE; PLE, Palatinose; TyrA, Tyrosine Arylamidase; URE, UREASE; dSOR, D-Sorbitol; SAC, Saccharose/Sucrose; dTAG, D-Tagatose; dTRE, D-Trehalose; CIT, Citrate (Sodium); MNT, Malonate; 5 KG, 5-Keto-D-Gluconate; lLATk, L-Lactate alkalinization; AGLU, ALPHA-Glucosidase; SUCT, Succinate alkalinization; NAGA, Beta-N-Acetyl-Galactosaminidase; AGAL, Alpha-Galactosidase; Phosphatase; GlyA, Glycine Arylamidase; ODC, Ornithine Decarboxylase; LDC, Lysine Decarboxylase; lHISa, L-Histidine assimilation; CMT, Coumarate; BGUR, BETA-Glucuronidase; O129R, O/129 RESISTANCE (comp.*Vibrio*.); GGAA, Glu-Gly-Arg-Arylamidase; lMLTa, L-MALATE assimilation; ELLM, Ellman; lLATa, L-Lactate assimilation.

**Table 2. Motility testing in a semisolid [SIM- Sulphide Indole Motility] media.**

| Strains | After 24 hrs | | | After 48 hrs | | |
|---|---|---|---|---|---|---|
| | H₂S | Indole | Motility | H₂S | Indole | Motility |
| *S.* Typhimurium ATCC14028 | + | – | + | + | – | + |
| NSP30218 | + | – | – | + | – | + |
| NSP30228 | + | – | – | + | – | + |
| NSP30229 | + | – | – | + | – | + |

Key: $H_2S$: Hydrogen Sulfide production, -: negative, +: positive. Typed strain *Salmonella* Typhimurium ATCC 14028 was used as control.

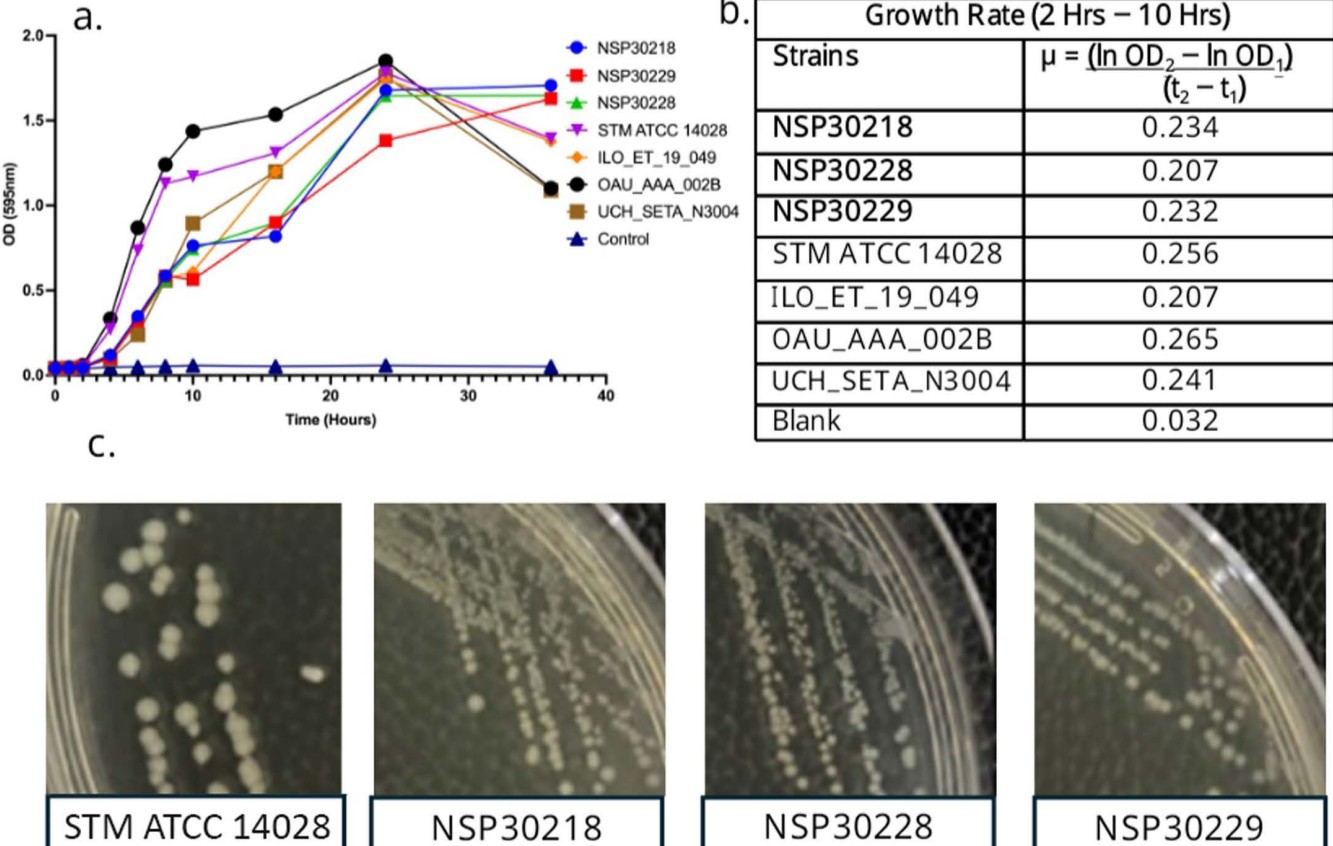

**Fig 3. Growth Rate of *Salmonella* Typhi strains. A.** Bacterial growth curve of slow-growing and typical *S.* Typhi strains in Dulbecco's Modified Eagle Medium (DMEM) broth at different time points using 595nm absorbance. **B.** Calculated growth rate of isolates where: µ = bacterial growth rate; t1 (start of log phase) = 2 hrs; t2 (end of log phase) = 16hrs; OD1 = optical density at 2 hrs; OD2 = optical density at 16 hrs. The three slow-growing *S.* Typhi strains had the lowest growth rate. **C.** Side by side colony size comparison of each slow-growing *S.* Typhi isolates with *S.* Typhimurium ATCC14028.

## *Salmonella* Sequence Types (ST) and phylogeny

SISTR software was used to predict the serovars of the three Typhi strains originally mis-identified as Gallinarum. All three belonged to ST2 and the common West African genotype 3.1.1., frequently recovered in Nigeria (Fig 2). Two of the three strains, from the same household, cluster together and had no SNP difference while the third isolate differed from them by

14 SNPs, within the range of SNP differences seen for 34 strains from outside the cluster (S2 Table). All three isolates did not have the most common VITEK biological profile [bionumber] for 3.1.1. Typhi strains (0005610440104210) and had bionumbers that differed from all other 3.1.1 strains recovered in the region (Fig 2). However, both the WGS and biochemical data suggest that their slow growth cannot be attributed to a common ancestor and that they represent independent *S.* Typhi lineages descended from within the most common local clade.

## Genome structures

We combined long- and short-read whole genome sequences to produce high-quality hybrid assemblies which were interrogated for the arrangement of the seven ribosomal RNA-flanked fragments in the *S.* Typhi genome, using socru, a purpose-built tool for determining Typhi genome structure. As shown in Fig 4, the arrangement of fragments in all three genomes was different from reference genome *Salmonella enterica* serovar Typhi CT18. Interestingly, however, two of the strains, NSP30228 and NSP30229 had arrangements that offset the origin of replication from the terminus by 20 or more. This arrangement was also seen in earlier isolate UCH_SETA_N3004 a previously isolated strain which, as can be seen from Fig 3 also grows somewhat slowly. The third *S.* Typhi isolate from the current study presented an arrangement which, while it did not show imbalance, has not been previously isolated.

## Discussion

Slow-growing, blood culture-derived *S.* Typhi were mis-identified as a cluster of *S.* Gallinarum in a clinical diagnostic laboratory based on acceptable and commonly-used methodology. Because *S.* Gallinarum is associated with poultry, and unusual in our setting, a zoonotic outbreak was initially presumed. In this study, we confirmed the isolates as Typhi and found that the API 20E misidentification stemmed from slow growth and consequent false negatives on key biochemical tests as well as slow motility. We further showed that they are not descended from a common slow-growing ancestor, but that slow growth could be attributed to independent chromosomal rearrangements.

Two of the individuals from which *Salmonella* was cultured in this study yielded genetically indistinguishable slow-growing *S.* Typhi, suggesting a point-source infection that was confirmed as such. The third isolate from a distant location differed by 14 SNPs, pointing to an independent infection event. However, because of temporal linkage, this infection was presumed part of the cluster until ruled out by whole genome sequencing. But for reference lab intervention, the most likely action would have been slaughter and discard of backyard poultry birds in the household with two patients. Given that the Ibadan municipality cannot offer compensation, this would have had severe economic consequences and might also have disrupted access to nutrients for the household, particularly protein, needed to boost immunity against typhoid. Given the identification of a third case from a different household, failing to confirm that the cluster was not *S.* Gallinarum may also have been followed by costly investigations to rule out human-to-human transmission after a presumed poultry-to-human transmission that never occurred. Thus, this study represents a case of judicious use of next generation sequencing that conserved economic household and health system resources.

The underlying reason why the isolates, and select others previously isolated in Ibadan, grew slowly was not immediately apparent. We initially suspected different lineages or a housekeeping gene polymorphism. These were however ruled out by lineage typing, SNP and phylogenetic analysis, and by biotyping. Waters et al [2022] [28] recently demonstrated that chromosomal rearrangements at Typhi rRNA operons could produce configurations that result in slow-growing *Salmonella*. They reported that rearrangements at the ribosomal RNA operons that result in configurations with over and under 50% of the genome being located at either side of the region between the origin of replication and replication terminus result in slow growth. Laboratory studies by different groups have determined that imbalances on either side of *ori* and *ter*, which can occur during prolonged growth in the laboratory can lead to fitness reductions, however their occurrence and significance in nature is unknown [28,29]. If they do occur, chromosomal rearrangements could go unrecognized even in locations where WGS is conducted because they can only be detected through long-read sequencing. In this study we

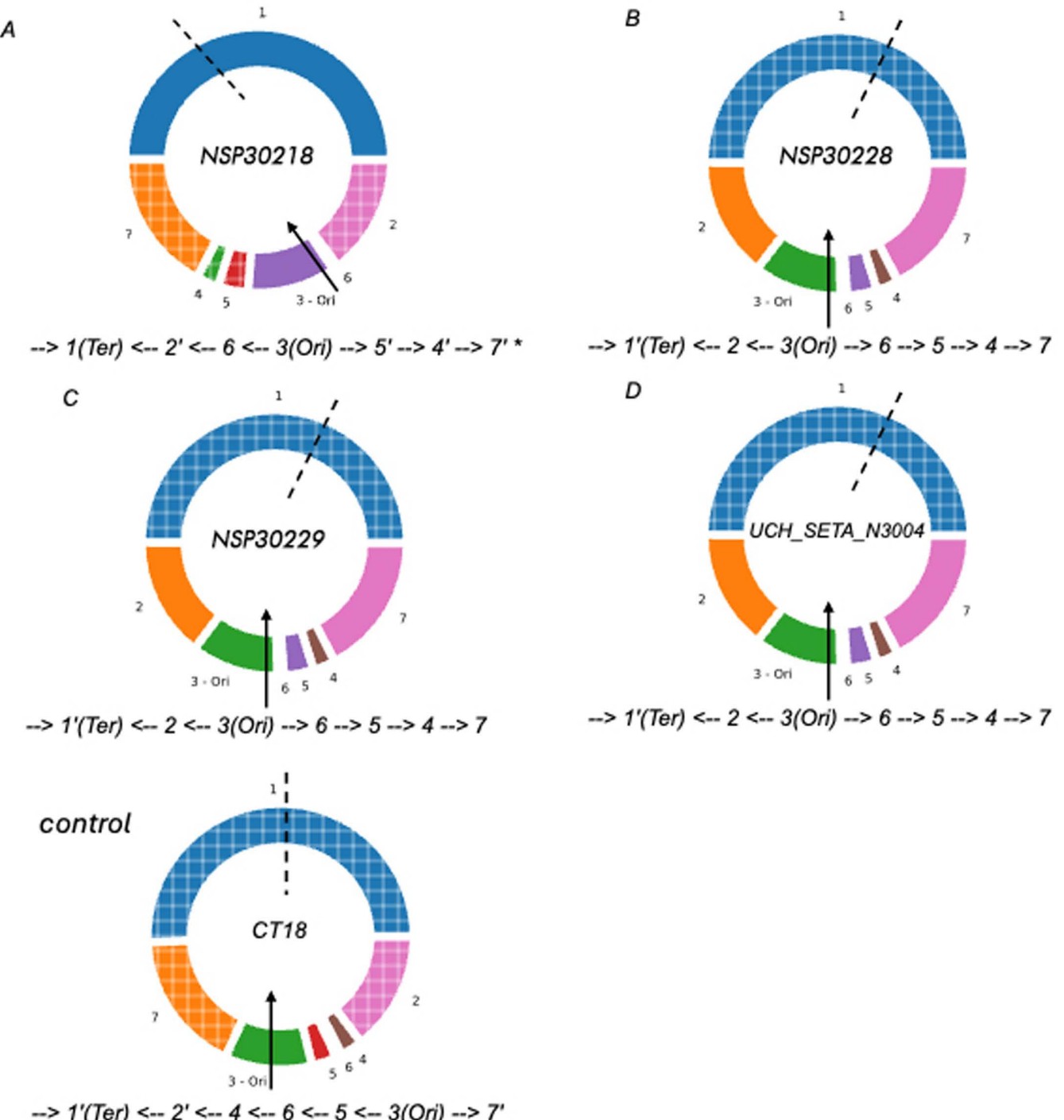

**Fig 4. Genome rearrangements of variants compared to CT18.** The origin of replication (oriC) is marked with an arrow in fragment 3 and the orientation of this fragment is fixed to match the orientation in the database reference, while the terminus of replication [*ter*] is marked with the dashed line in fragment 1. The direction and arrangement of fragments are indicated by the notations below each genome. A prime indicates the fragment [not the operon] is inverted. A (strain NSP30218) has a novel genome structure indicated by the * but oriC is nearly opposite ter so that the genome is balanced, as is that of control strain *S.* Typhi CT18, while B, C and D all have similar genome structures.

used hybrid assemblies study genome fragment arrangements and found that the two of the three slow-growing *Salmonella* strains from this study, and another archival strain, indeed had unbalanced chromosomal configurations compared to other Typhi. Their slow growth in turn resulted in incomplete biochemical profiles being taken at 24 hours with the API system, and slow motility that could not be detected by hanging drop method. The consequent misidentification as less biochemically active, non-motile serovar Gallinarum precluded Vi antisera confirmation, which is expensive in our setting and reserved for suspected Typhi only. The third isolate had a novel configuration but was not unbalanced. While we confirmed that it is a slow-grower, the basis for its slow growth remains unclear.

This report is associated with a few limitations. As only three slow-growing isolate were detected and reported, there are limits on the generalizability of our conclusions. While we identified chromosomal rearrangements in two isolates, this small number prevents us from establishing the frequency of such events in the broader *S.* Typhi population. Future studies with larger numbers of slow-growing *S.* Typhi could provide more robust evidence regarding the prevalence of these chromosomal rearrangements and their clinical significance. We do not know whether such isolates can be obtained in geographical locations outside Ibadan, Nigeria, and can only hypothesize that they may occur in endemic areas with similar epidemiological characteristics. Hybrid assemblies of whole-genome sequenced isolates from multi-center studies or across different geographical regions would provide valuable insights into whether similar slow growing isolates occur elsewhere and whether the chromosomal rearrangements we observed are widespread or regionally confined. We recommend that human isolates of slow-growing *S.* Typhi or presumptive *S.* Gallinarum be prioritized for long-read sequencing.

Another limitation is that our study did not explore the potential clinical implications of these slow-growing variants, including whether they cause different disease manifestations or treatment responses compared to typical *S.* Typhi strains. The selective pressures that might encourage or discourage these chromosomal configurations in clinical settings also remain unexplored. Despite these limitations, our findings highlight an important diagnostic challenge that may have broader implications for typhoid surveillance, particularly in resource-limited settings where API 20E and similar biochemical identification systems are commonly used.

We have shown that slow growing Typhi can be mistaken for Gallinarum in typhoid endemic locales. In future, we recommend that in situations where Gallinarum is rarely or never reported, suspected Gallinarum be sent for reference laboratory evaluation, as in this case, or processed for re-identification at an extended timeline, which we confirmed produced the correct identification on API. Our data has also shown that Typhi with rearrangements that produce a growth disadvantage circulate in our endemic setting, opening the question about possible selective advantages of unbalanced chromosomal profiles.

## Conclusion

*S.* Typhi and *S.* Gallinarum are biochemically similar group D1 serovars with similar API20E profiles. Mis-identification of slow-growing *S.* Typhi as *S.* Gallinarum is therefore possible, as occurred in this study. As many resource-limited laboratories may not have anti-Gallinarum sera on site, suspected *S.* Gallinarum isolates in typhoid endemic areas should be evaluated for motility in semi-solid media after extended incubation and verified by serological or molecular methods. More research needs to be conducted to determine the prevalence and evolutionary rationale for *S.* Typhi that grow slowly due to unbalanced chromosomal configurations that arise from rearrangements at *rrn* operons.

## Supporting information

**S1 Table. Sources and properties of *Salmonella* isolates used in this study.**
(XLSX)

**S2 Table. Matrix showing SNP differences among strains included in this study.**
(XLSX)

## Acknowledgments

We thank SETA-Nigeria program officer Ifiok Udofia, data officer Oluwaseun Alaran for providing metadata and Miss Elizabeth T Akande, Mr. Ezekiel Jacobs, Mr Oluwaseun Alaran and Miss Priscilla Bamidele for technical assistance. We are grateful to Gemma Langridge for helpful discussions.

## Author contributions

**Conceptualization:** Veronica O. Ogunleye, Iruka N. Okeke.

**Data curation:** Gabriel Temitope Sunmonu, Odion O. Ikhimiukor, Iruka N. Okeke.

**Formal analysis:** Gabriel Temitope Sunmonu, Odion O. Ikhimiukor, Precious E. Osadebamwen.

**Funding acquisition:** Aderemi Kehinde, Iruka N. Okeke.

**Investigation:** Gabriel Temitope Sunmonu, Veronica O. Ogunleye, Odion O. Ikhimiukor, Oluwafemi A. Popoola, Precious E. Osadebamwen, Iruka N. Okeke.

**Methodology:** Odion O. Ikhimiukor, Oluwafemi A. Popoola, Precious E. Osadebamwen, Iruka N. Okeke.

**Project administration:** Veronica O. Ogunleye, Oluwafemi A. Popoola, Aderemi Kehinde, Iruka N. Okeke.

**Supervision:** Oluwafemi A. Popoola, Aderemi Kehinde, Iruka N. Okeke.

**Validation:** Gabriel Temitope Sunmonu, Odion O. Ikhimiukor, Iruka N. Okeke.

**Visualization:** Gabriel Temitope Sunmonu, Odion O. Ikhimiukor, Precious E. Osadebamwen.

**Writing – original draft:** Gabriel Temitope Sunmonu, Iruka N. Okeke.

**Writing – review & editing:** Veronica O. Ogunleye, Odion O. Ikhimiukor, Oluwafemi A. Popoola, Precious E. Osadebamwen, Iruka N. Okeke.

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
