## [Decision Letter · Decision Letter 0]

PGPH-D-25-00768

Slow-growing Salmonella enterica Typhi mis-identified as Salmonella Gallinarum in Ibadan, Nigeria

Dear Dr. Okeke,

Thank you for submitting your manuscript to PLOS Global Public Health. After careful consideration, we feel that it has merit but does not fully meet PLOS Global Public Health’s publication criteria as it currently stands. Therefore, we invite you to submit a revised version of the manuscript that addresses the points raised during the review process.

We look forward to receiving your revised manuscript.

Kind regards,

Mila Shakya

Academic Editor

Journal Requirements:

Additional Editor Comments (if provided):

Reviewers' comments:

Reviewer's Responses to Questions

**Comments to the Author**

1. Does this manuscript meet PLOS Global Public Health’s publication criteria?

Reviewer #1: Yes

Reviewer #2: Yes

2. Has the statistical analysis been performed appropriately and rigorously?

Reviewer #1: Yes

Reviewer #2: Yes

3. Have the authors made all data underlying the findings in their manuscript fully available (please refer to the Data Availability Statement at the start of the manuscript PDF file)?

Reviewer #1: Yes

Reviewer #2: No

4. Is the manuscript presented in an intelligible fashion and written in standard English?

Reviewer #1: Yes

Reviewer #2: Yes

Reviewer #1: This well-designed study documents three S. Typhi bloodstream isolates that were erroneously called S. Gallinarum in a routine clinical laboratory, then re-typed by biochemical retesting, Illumina + Nanopore sequencing, and genomic analysis. The work has clear public-health relevance for typhoid-endemic, resource-limited settings and illustrates how rrn-mediated chromosomal rearrangements can generate slow-growing S. Typhi with atypical phenotypes.

Major comments

1. Figure 4 has poor resolution. Please supply individual high-resolution (≥300 dpi) TIFF/PNG files. It is impossible to read the maps.

2. The Discussion would benefit from limitations paragraph acknowledging (i) the very small number of cases and (ii) potential geographic specificity; please consider other limitations.

3. Only a single p=0.001 Fisher test is reported (GGT prevalence). Please specify software (e.g., R v4.3), exact two-sided/one-sided test, and numerator/denominator to aid reproducibility.

Minor comments:

line 47: import -> importance, line 47, 67 and others.... Salmonella should be italic

Reviewer #2: The authors presented an interesting case of using NGS to correctly detect slow-growing serovar Typhi that were previously misidentified as serovar Gallinarum. There are a few comments that I would recommend to the authors to consider:

1. The Biosamples of the 3 isolates do not exist on NCBI. The authors should look into making this rectified

2. In the abstract, the first mention of S. Gallinarum should be Salmonella enetrica subsp. enterica Gallinarum. Also it shouldn't be abbreviated as S. Gallinarum but preferably as serovar Gallinarum given that is a serovar and not a species name of Salmonella. Kindly correct throughout the manuscript. The same applies to S. Typhi. It should be corrected to serovar Typhi.

3. The introduction could use some historical context and more information on the different serovars of S. enterica and their public health implications. The Gallinarum serovar is usually associated with poultry and the authors should also address its risk as a zoonotic pathogen

5. Line 171 Gallinarum not GAllinarum

6. Figure 1, what does INTS signify? The figure caption should be self explanatory without reading the article

7. Figure 4 is hard to read a more legible Figure would be appropriate

**Do you want your identity to be public for this peer review?** For information about this choice, including consent withdrawal, please see our Privacy Policy

Reviewer #1: No

Reviewer #2: No

---

## [Editor Report · Decision Letter 1]

PGPH-D-25-00768R1

Slow-growing Salmonella enterica serovariety Typhi mis-identified as serovariety Gallinarum in Ibadan, Nigeria

Dear Dr. Okeke,

Thank you for submitting your manuscript to PLOS Global Public Health. After careful consideration, we feel that it has merit but does not fully meet PLOS Global Public Health’s publication criteria as it currently stands. Therefore, we invite you to submit a revised version of the manuscript that addresses the points raised during the review process.

We look forward to receiving your revised manuscript.

Kind regards,

Mila Shakya

Academic Editor

Journal Requirements:

Additional Editor Comments (if provided):

Thank you for your revised manuscript submission. However, the track-changed version provided does not show all tracked changes. Could you please resubmit a version of the manuscript with all changes clearly marked using Word's Track Changes feature?

The figures provided are still not sufficiently clear for publication. We kindly request that you submit improved versions of all figures with higher resolution and enhanced clarity.
---

## [Editor Report · Decision Letter 2]

Slow-growing Salmonella enterica serovariety Typhi mis-identified as serovariety Gallinarum in Ibadan, Nigeria

PGPH-D-25-00768R2

Dear Prof Okeke,

We are pleased to inform you that your manuscript 'Slow-growing Salmonella enterica serovariety Typhi mis-identified as serovariety Gallinarum in Ibadan, Nigeria' has been provisionally accepted for publication in PLOS Global Public Health.

Best regards,

Mila Shakya

Academic Editor